# Drip irrigation and sulphur fertilization influenced fodder yield, quality and water use efficiency of groundnut in arid region

**Priyanka Gautam**[1,2]*, **S. R. Bhunia**[1], **A. Sahoo**[2], **R. K. Sawal**[2], **Shantanu Rakshit**[2], **V. K. Yadav**[2], **B. Lal**[3], **Ramniwas**[1], **Gograj**[1], **Rajesh Bishnoi**[3], **V. S. Rathore**[4]

1 Department of Agronomy, College of Agriculture, SKRAU, Bikaner, India, 2 ICAR-National Research Centre on Camel, Bikaner, India, 3 ICAR-Indian Institute of Pulses Research, Regional Research Centre, Bikaner, India, 4 ICAR-Central Arid Zone Research institute, Regional Research Station, Bikaner, India

* priyanakagautam@gmail.com

**Data Availability Statement:** All relevant data are within the paper.

**Funding:** The author(s) received no specific funding for this work.

## Abstract

Availability of ample and nutritious fodder for livestock is always a challenge in arid region. Choice of crops such as groundnut that can fulfil the requirement of fodder with its crop residues along with human needs can be a viable option to bridge the gap between availability and requirement of fodder. The fodder yield and quality largely depend on soil moisture and nutrient supply especially sulphur (S), a key nutrient for improving groundnut fodder quality. However, no researchers have given emphasis on coupling effect of drip irrigation (DI) and sulphur on fodder yield, quality, digestibility and water use efficiency (WUE). Therefore, the study was conducted to determine the effects of different regimes of DI and S on productivity and quality of fodder. Results revealed that higher regimes of DI i.e. 0.8 +1.0 PE(pan evaporation) and 1.0 PE level of irrigation along with 40 kg S ha$^{-1}$ significantly improved the yield, primary quality traits (crude protein, ether extract and ash), digestibility indices and significant reduction in Fibers which indicates improvement in quality of fodder. Fodder productivity was 27.0 and 25.6% higher in 1.0 PE and 0.8+1.0 PE level of irrigation, respectively, as compared to 0.6 PE level (lower water regime) of irrigation, although 0.6 PE level of irrigation recorded higher WUE and was at par with 0.8 PE and 0.6 +1.0 PE level of DI. By changing the levels of DI from 1.0 PE to 0.8+1.0 PE, considerable water can be saved without affecting the yield and quality of fodder. Similarly, crop responded to S up to 60 kg ha$^{-1}$ but at par with 40 kg S ha$^{-1}$ indicating that application of extra S after 40 kg did not warrant any extra benefit in terms of fodder yield, WUE and quality of fodder. Thus, adjusting the PE levels of DI for water saving and optimal S application can be a sustainable strategy to improve the productivity and quality of groundnut fodder in arid region.

## Introduction

Rapid increase in human population imposing huge pressure on livestock to fulfil the need for animal products. Fodder quality is also essential with quantity for higher productivity of livestock [1]. Fodder deficit has to be bridge up either by enhancing productivity, utilizing

**Competing interests:** The authors have declared that no competing interests exist.

unexploited feed resources like crop residues of various crops, or by horizontal expansion of area under forage crop, which has limited scope [2]. In arid regions, livestock rearing is difficult due to limited supply of forages because of water scarcity and poor soil fertility [3]. Therefore, a serious strategy is essential for continuous fodder supply by forage cultivation and feed for livestock [4] in these areas. Forage quality (palatability, digestibility, intake, nutrient content) is also need to be maintained or improved along with yield for better livestock productivity. Forage quality is affected by several factors such as harvesting time, maturity age of forage, type of forage species, prevailing weather and crop management practices [5]. Competition for natural resources especially water for production of food and forage has put arid regions under pressure. Dual purpose crops such as groundnut needs to be cultivated, which is a valuable source of nutrient, and every part of groundnut has commercial value [6]. The kernels are good source of quality oil and protein for humans, green leaves, stems and shells of the pods are used as animal feed, its haulm is considered as good dry fodder for animals, cake obtained after oil extraction is also used as animal feed, especially in arid regions [7], where mitigation of fodder requirement is always a challenge.

Scarcity of water and nutrients, poor soil fertility with fragile ecosystem are the major characteristics of arid regions. Crop failure is quite common in this region and the reasons may be unpredictable weather, erratic and uneven rainfall leading to prolonged dry spell (may be up to > 30 days)and high salinity [8]. In addition to the harsh climate, inefficient utilization of existing limited resources under arid climate is also a serious threat to plant productivity [9]. From last few years' irrigation facility improved in the region and it enhanced crop productivity, however, overutilization of groundwater for irrigation, unreasonable land use and other inappropriate agronomic measures [10] lead to salinization and deterioration of soil health [11]. There is a large gap exists between rainfall and potential evapotranspiration that indicates the essentiality of irrigation for meeting water requirement of crops in this region. Thus, the importance of agro-management practices that preserve water has increased [12]. Ground nut farmers usually apply excessive irrigation i.e. around 600–700 mm water to maximize yield [9]. This poor and inefficient management of irrigation water resulted in reduction of WUE, and economic benefits along with several environmental issues. Therefore, suitable irrigation regimes like drip irrigation are needed for augmenting water use efficiency for sustainable peanut production in arid regions of India. Drip irrigation promises complete elimination of the problem of water stress even under severe water scarcity condition. Unlike the conventional method of irrigation, pipe network and emitters in drip irrigation delivers the water near the root zone of crops without much loss of water, resulting in higher water productivity [13].

Another important factor for improving crop production and productivity after irrigation is through modifying soil nutrient supply. Sulphur's essentiality for plant growth and development has also been acknowledged for improving crop productivity [14], quality [15], and plants' abiotic stress responses [16]. Therefore, access to an adequate supply of S for plants throughout their development is necessary for optimum crop performance [17]. Despite all this, S has received little attention for many years, until only recently [18]. This is largely because previously it was a thought that fertilizers and atmospheric deposition adequately supplied the soil with enough sulphur [19]. Subsequently, due to the importance of S nutrient in plant functions (such as in sugar production, carbon dioxide assimilation, nitrogen (N) fixation and protein formation), S is increasingly becoming more important [20]. Notably, scientific research indicates that the farmers will have to start applying S through fertilization to achieve full potential of crop in terms of yield, quality and to make efficient use of applied N for protein and enzyme synthesis [16, 19]. In most of the groundnut growing tracts, the level of available sulphur reaches below the critical limit and groundnut crop is bound to suffer on account of sulphur deficiency [21]. Since groundnut is rich in both oil and protein, the

requirement of S for this crop is substantial [21]. Therefore, the present investigation was planned with the hypothesis that drip irrigation at suitable pan evaporation level can save enough amount of water and optimum sulphur fertilization can improve forage productivity and quality of groundnut in hot arid region. Accordingly, this study was conducted to determine the effect of drip irrigation levels and different doses of sulphur in groundnut on fodder yield, fodder value, quality of dry fodder and water saving in arid environments.

## Materials and methods

### Experimental site and climatic conditions

The field experiment was conducted during kharif season of two consecutive years of 2020 and 2021 at Experimental Farm, College of Agriculture, S.K. Rajasthan Agricultural University, Bikaner (Rajasthan), situated at 28° 10´N latitude and 73° 35´ E longitude at an altitude of 235 meters above mean sea level. Bikaner falls under arid climate with average annual rainfall of about 265 mm, receiving more than 80 per cent in the *kharif* season (July-September) through South West monsoon. During summer, the maximum temperature may go as high as 48° C while in the winters it may fall as low as 0°C. The weather data was obtained from Meteorological observatory of Agricultural Research Station, Bikaner and presented as Fig 1. The maximum and minimum temperature ranged between 26.6°C to 43.2°C and 8.5°C to 31.1°C during the crop growing season of 2020 and 32.1°C to 41.1°C and 12.0°C to 27.9°C during the crop growing season of 2021, respectively. Crop received 159.9 mm and 251.2 mm of rainfall with 12 and 20 rainy days in the growing season of 2020 and 2021, respectively. The soil of the experimental field was loamy sand in texture and slightly alkaline in reaction (pH 8.4), poor in organic carbon (0.11%), low in available nitrogen (104 kg ha$^{-1}$) but medium in available phosphorus (14 kg ha$^{-1}$), potassium (75 kg ha$^{-1}$).

### Experimental design and treatments

The experiment was laid out in split plot design with irrigation as main plot treatment and sulphur fertilization as sup-plot treatment and replicated thrice. Six levels of drip irrigation viz., 0.6 PE(from sowing to maturity), 0.8 PE(from sowing to maturity), 1.0 PE (from sowing to maturity), 0.6 PE (0-45DAS) + 0.8 PE (46 DAS to maturity), 0.6 PE (0-45DAS) + 1.0 PE (46 DAS to maturity) and 0.8 PE (0-45DAS) + 1.0 PE (46 DAS to maturity) and four levels of sulphur fertilization i.e., 0, 20, 40 and 60 kg S ha$^{-1}$ were used. Wettable powder of soluble sulphur was applied as per the treatments, it is a readily available form of S, and therefore it was applied at the time of sowing. The layout of the experiment was same during both the years. For field preparation, one pre-sowing irrigation (*palewa*) was applied and ploughing was done by tractor drawn Rotavator. Thereafter, the field was laid out manually into plots according to the plan of work with the provision of path. Irrigations were applied as per treatments.

### Crop management

The groundnut seed was treated before sowing adopting FIR (seed was treated in the sequence of fungicide, insecticide, Rhizobium) method to ensure protection from soil borne diseases and for enhancing nitrogen fixationwith fungicide Bavistin @ 2 g kg$^{-1}$, imidaclorpid @ 3 ml kg$^{-1}$ and *Rhizobium*. The groundnut variety HNG-123 was sown manually by kera method (dropping seeds in furrows followed by covering with soil)at depth of 5–6 cmat a seed rate of 100 kg ha$^{-1}$. For fertilizer management, 40 kg N ha$^{-1}$, 40 kg P$_2$O$_5$ ha$^{-1}$, 60 kg K$_2$O ha$^{-1}$were given

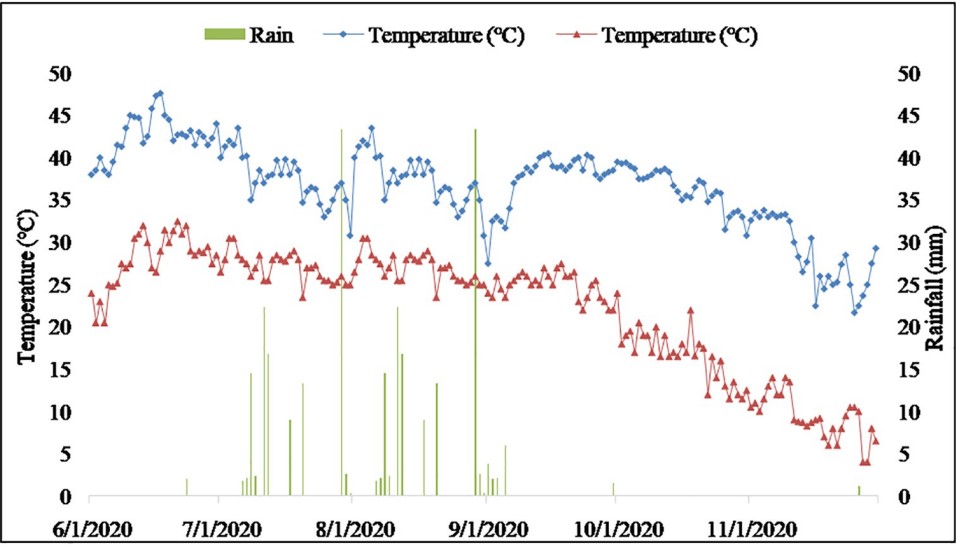

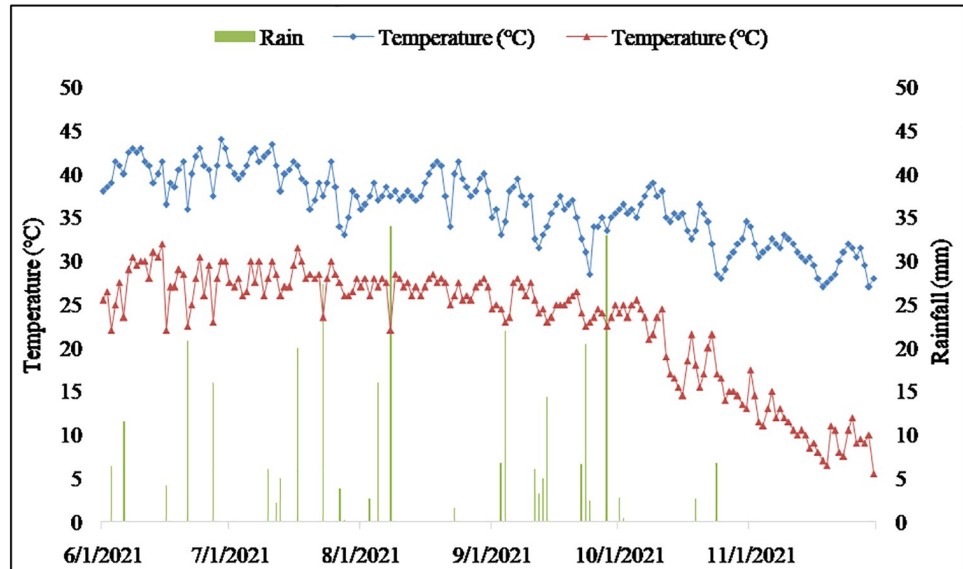

**Fig 1. Maximum and minimum temperature, rainfall recorded daily during the crop growth period i.e. from June to November in 2020 and 2021.**

through Urea, DAP and MOP. Nitrogen, phosphorus, potassium and sulphur (as per the treatments) were applied as basal before sowing. Prophylactic plant protection measures were undertaken to protect the crop from insects and diseases. Two sprays of Streptocycline @ 0.5 g liter$^{-1}$ + Copper oxy chloride @ 2.5 g liter$^{-1}$ +Mencozeb @ 2 g liter$^{-1}$ of water at 80 DAS and 100 DAS were done for controlling collar rot and blight disease.

## Irrigation water application

Irrigation was scheduled based on pan evaporation. Irrigations through drip were scheduled on alternate days as per treatment. The quantity of water was calculated as follows:

$$Irrigation\ water(mm) = PE \times Irrigations\ levels$$

Where,

$$PE = Pan\ evaporation(mm)$$

The lateral drip lines are laid on the soil surface at a line-to-line distance of 100 cm and dripper to dripper distance was 30 cm with 4 lit hr$^{-1}$ discharge.

## Fodder yield

Groundnut was harvested manually and after removing the pods of the plants from net plot area, biomass was recorded from each treatment and green and dry fodder productivity was recordedas t ha$^{-1}$.

## Fodder quality parameters

The plant samples were collected at the time of harvesting for analysis of fodder quality parameters. Contents of ash (%), crude protein (CP; %) and ether extract (EE; %) was estimated as per AOAC [22]. Neutral and acid detergent fiber (NDF and ADF) were determined as per Van Soest et al. [23] while lignin was analyzed by procedure of Robertson and Van Soest [24]. The haulm yield was multiplied with the content of CP, EE and ash% for estimation of respective yield under each treatment.

Secondary quality parameters also determined such as DMI, TDN, DMD, NE$_L$, RFV, RFQ, DE, ME and NE. Dry matter intake (DMI), total digestible nutrients (TDN), dry matter digestibility (DMD), net energy for lactation (NEL) were determined by the equations suggested by Horrocks and Vallentine [25].

$$DMI(\%) = \frac{120}{NDF}$$

$$TDN(\%) = -1.291 \times ADF + 101.35$$

$$DMD(\%) = 88.9 - (0.779 \times ADF)$$

$$NE_L(Mcal\ kg^{-1}) = [1.044 - (0.0119 \times ADF)] \times 2.205$$

$$NE_L(MJ\ kg^{-1}) = NE_L(Mcal\ kg^{-1}) \times 4.184$$

Relative feed value (RFV) is an important indicator to estimate the digestible dry matter from ADF and calculates the dry matter intake potential from NDF. Higher the RFV indicates higher quality of fodder determined by the equation given by Horrocks and Vallentine [25].

$$RFV = \frac{DMI(\%) \times DMD(\%)}{1.29}$$

Relative feed quality (RFQ) index reflects the Fiber digestibility to estimate intake as well as total digestible nutrients substitutes for DDM. Fodder/feed containing higher NDF, make RFQ, a better predictor of fodder quality than RFV. The RFQ emphasizes on digestibility of Fiber while RFV uses DDMI [26].

$$RFQ = \frac{DMI(\%) \times TDN(\%)}{1.23}$$

Digestible energy (DE) provides an indication of actual amount of energy from a feed/

fodder that can be used by animal and was estimated by the formula of Fonnesbeck et al. [27].

$$DE(Mcal\,kg^{-1}) = 0.27 + [0.0428 \times DMD(\%)]$$

$$DE(MJ\,kg^{-1}) = DE(Mcal\,kg^{-1}) \times 4.184$$

Metabolizable energy (ME) refers to the digestible energy minus energy lost in urine plus energy lost in the form of gaseous production of methane by rumen and hind gut microbes during digestion.

$$ME(MJ\,kg^{-1}) = DE(MJ\,kg^{-1}) \times 0.821$$

$$\text{Digestible feed energy}(\text{DFE}) = \{\frac{4.4 \times TDN(\%)}{1.23100}\} \times 4.184$$

Net energy (NE) was calculated as difference in energy lost from excreta and heat produced in digestive and metabolic processes [28].

$$NE(MJ\,kg - 1) = \left[\frac{\{TDN(\%) \times 3.65\} - 100}{188.3} \times 6.9\right.$$

## Nutrient analysis

The representative samples of haulm/dry fodder were taken at the time of threshing were thoroughly ground to pass through 40mesh sieve and analyzed for nitrogen, and sulphur content. Nitrogenand sulphur content was estimated by procedures of Subbiah and Asija [29] and Chesnin and Yien [30], respectively.

## Water use and water use efficiency (WUE)

Total water applied to the field was calculated at different PE levels viz., 0.6 PE, 0.8 PE, 1.0 PE and 0.6+0.8 PE, 0.6+1.0 PE and 0.8+1.0 PE. The amount of water received through rainfall during crop growing period was also added in the estimation of water use. Water use efficiency was calculated as the ratio of groundnut green and dry fodder yield to total water used in the particular treatment and expressed in kg ha$^{-1}$ mm.

$$WUE(kg\,ha^{-1}mm) = \frac{Fodder\,yield(kg\,ha^{-1})}{wateruse(mm)}$$

## Statistical analysis

The data of fodder yield, quality parameters, digestibility indices, nutrient uptake and WUE were recorded and analyzed in analysis of variance (ANOVA) for split-plot design in excel. All the recorded data is distributed normally with equal variances. Treatment significance was determined using the F-test, and comparisons were made by using critical difference (CD) at the 5% level of significance. Regression analysis was performed between fodder quality parameters and irrigation water and sulphur regimes.

## Results

### Effect on fodder productivity of groundnut

Green and dry fodder yields of groundnut significantly influenced with different treatments of drip irrigation levels and S fertilization during both the years (Table 1). Maximum fodder yield was recorded in 1.0 PE level of drip irrigation but it was at par with 0.8 +1.0 PE drip irrigation,

**Table 1. Effect of drip irrigation and sulphur levels on green and dry fodder yield of groundnut during 2020 and 2021.**

| Drip irrigation levels (PE) | Sulphur levels (kg ha$^{-1}$) | | | | | | | |
|---|---|---|---|---|---|---|---|---|
| | 0 | 20 | 40 | 60 | 0 | 20 | 40 | 60 |
| | 2020 | | | | 2021 | | | |
| **Green fodder yield (t ha$^{-1}$)** | | | | | | | | |
| 0.6 | 10.70c | 11.32b | 13.06b | 13.25b | 11.46c | 12.08c | 13.63c | 14.03b |
| 0.8 | 14.69ab | 14.77a | 14.18b | 16.12a | 15.47ab | 15.47ab | 14.88c | 16.82a |
| 1.0 | 16.29a | 17.00a | 17.65a | 17.40a | 17.03a | 17.73a | 18.39a | 18.15a |
| 0.6 + 0.8 | 13.22b | 15.23a | 15.57ab | 13.46b | 13.31bc | 15.32b | 15.66bc | 13.55b |
| 0.6 + 1.0 | 14.29ab | 15.17a | 15.63ab | 16.51a | 14.42b | 15.29b | 15.73bc | 16.50a |
| 0.8 + 1.0 | 16.16a | 17.01a | 16.67a | 17.03a | 16.84a | 17.69a | 17.34ab | 17.70a |
| S.Em.± | 0.86 | | | | 0.80 | | | |
| CD$_{P=0.05}$ | 2.47 | | | | 2.30 | | | |
| **Dry fodder yield (t ha$^{-1}$)** | | | | | | | | |
| 0.6 | 3.43c | 3.63b | 4.19b | 4.25b | 3.65c | 3.85c | 4.34c | 4.47b |
| 0.8 | 4.54ab | 4.57a | 4.38b | 4.98a | 4.78ab | 4.78ab | 4.59c | 5.19a |
| 1.0 | 4.71a | 4.92a | 5.11a | 5.03a | 4.92a | 5.13a | 5.32a | 5.25a |
| 0.6 + 0.8 | 3.95b | 4.55a | 4.65ab | 4.02b | 3.97bc | 4.57b | 4.67bc | 4.04b |
| 0.6 + 1.0 | 4.24ab | 4.50a | 4.63ab | 4.89a | 4.27b | 4.53b | 4.66bc | 4.88a |
| 0.8 + 1.0 | 4.73a | 4.98a | 4.88a | 4.99a | 4.92a | 5.17a | 5.07ab | 5.18a |
| S.Em.± | 0.26 | | | | 0.24 | | | |
| CD$_{P=0.05}$ | 0.74 | | | | 0.69 | | | |

whereas lowest fodder yield was recorded in 0.6 PE level of irrigation. On an average, drip irrigation levels of higher PE, i.e., 1.0 PE and 0.8 +1.0 PE increased dry fodder yield by 1045, 292, 713 and 444 kg ha$^{-1}$ over 0.6 PE, 0.8PE, 0.6+0.8PE and 0.6+1.0PE level of drip irrigation, respectively. Fodder yield was 27.0 and 25.6% higher in 1.0 PE and 0.8+1.0 PE level of irrigation, respectively, as compared to 0.6 PE level of irrigation.

Maximum green and dry fodder yield was recorded with 60 kg S ha$^{-1}$ which was at par with 40 kg S ha$^{-1}$, while lowest fodder yield was recorded in control (Table 1). On an average, application of 20, 40 and 60 kg S ha$^{-1}$ resulted in 5.8, 8.4, 9.7% higher yield of groundnut, respectively, when computed over no S application. The increase in dry fodder yield was 253 kg ha$^{-1}$, when level of S was increased from 0–20 kg S ha$^{-1}$; this increase in yield was 111 kg ha$^{-1}$, when level of S increased from 20–40 kg S ha$^{-1}$, and then when level of S increased from 40–60 kg S ha$^{-1}$ yield increase was only 57 kg ha$^{-1}$. As per the interaction effect it was observed that drip irrigation level of 1.0 PE produced maximum fodder yield when supplied with 40 kg S ha$^{-1}$, although it was at par with the 1.0 PE and 60 kg S ha$^{-1}$, 0.8+1.0 PE and 60 kg S ha$^{-1}$, 0.8+1.0 PE and 40 kg S ha$^{-1}$.

## Quality parameters of groundnut fodder

**Crude protein, ether extract and ash content and yield.** Different levels of drip irrigation and sulphur fertilization significantly influenced the content and yield of CP, EE and ash in groundnut fodder during both the years (Tables 2 and 3). It can be seen that significantly highest content and yield of CP (16.98% and 8.57 q ha$^{-1}$), EE(3.50% and 1.76 q ha$^{-1}$) and ash(9.84% and 4.98 q ha$^{-1}$) was recorded in 1.0 PE level of drip irrigation, which was at par with 0.8 +1.0 PE level of drip irrigation. Whereas, lowest CP, EE and ash content and yield was recorded in 0.6 PE level of irrigation. On an average, the content and yield of CP, EE and ash was higher in 2021 over 2020. Application of higher doses of S(40 and 60 kg S ha$^{-1}$) contributed positively in

**Table 2. Effect of drip irrigation and sulphur levels on content and yield of crude protein (CP), ether extract (EE) and ash of groundnut fodder during 2020.**

| | Content (%) | | | Yield (q ha$^{-1}$) | | | Fiber (%) | | |
|---|---|---|---|---|---|---|---|---|---|
| | CP | EE | Ash | CP | EE | Ash | CF | NDF | ADF |
| **Drip irrigation levels (PE)** | | | | | | | | | |
| 0.6 | 14.14b | 2.84b | 6.76b | 5.43d | 1.11b | 2.65c | 25.32a | 38.53a | 29.83a |
| 0.8 | 15.39ab | 3.33ab | 9.26a | 7.05ab | 1.56a | 4.33ab | 21.25b | 32.50b | 23.98b |
| 1.0 | 16.79a | 3.49a | 9.83a | 8.30a | 1.72a | 4.88a | 20.28b | 30.90b | 22.52b |
| 0.6 + 0.8 | 14.79ab | 3.08ab | 8.61a | 6.41cd | 1.34ab | 3.71b | 23.63ab | 34.24ab | 26.31ab |
| 0.6 + 1.0 | 15.21ab | 3.31ab | 8.98a | 6.88bc | 1.50a | 4.13ab | 22.16ab | 33.16b | 24.89b |
| 0.8 + 1.0 | 16.25ab | 3.44ab | 9.44a | 7.91ab | 1.67a | 4.63a | 20.26b | 32.05b | 23.13b |
| S.Em.± | 0.68 | 0.20 | 0.39 | 0.43 | 0.12 | 0.27 | 1.09 | 1.39 | 1.28 |
| LSD$_{P=0.05}$ | 2.14 | 0.62 | 1.24 | 1.36 | 0.37 | 0.85 | 3.42 | 4.37 | 4.04 |
| **Sulphur levels (kg ha$^{-1}$)** | | | | | | | | | |
| 0 | 14.82b | 2.99b | 7.35d | 6.34b | 1.29c | 3.20c | 23.87a | 35.75a | 27.54a |
| 20 | 15.31ab | 3.25ab | 8.82c | 6.91ab | 1.47b | 4.04b | 22.27b | 34.48a | 25.27bc |
| 40 | 15.68ab | 3.29a | 9.17b | 7.28a | 1.53a | 4.29ab | 21.38b | 32.63b | 24.16cd |
| 60 | 15.90a | 3.47a | 9.91a | 7.45a | 1.63a | 4.68a | 21.08b | 31.40b | 23.46d |
| S.Em.± | 0.35 | 0.09 | 0.19 | 0.22 | 0.05 | 0.14 | 0.44 | 0.53 | 0.56 |
| LSD$_{P=0.05}$ | 0.99 | 0.27 | 0.55 | 0.63 | 0.15 | 0.41 | 1.27 | 1.52 | 1.60 |
| **I x S $_{P=0.05}$** | 2.43 | 0.66 | 1.36 | 1.55 | 0.38 | 1.00 | 3.10 | 3.72 | 3.91 |

better vegetative, and reproductive of the crop resulting in higher N content in haulm leading to better content and yield of CP, EE and ash in fodderduring both the years of 2020 and 2021 (Tables 2 and 3). Maximum content and yield of CP (16.08% and 7.66 q ha$^{-1}$), EE(3.47% and 1.66 q ha$^{-1}$) and ash(9.92% and 4.75 q ha$^{-1}$), which was at par with 40 kg S ha$^{-1}$, whereas lowest contents and yield was recorded in control. On an average, application of 20, 40 and 60 kg S

**Table 3. Effect of drip irrigation and sulphur levels on content and yield of crude protein (CP), ether extract (EE) and ash of groundnut fodder during 2021.**

| | Content (%) | | | Yield (q ha$^{-1}$) | | | Fiber (%) | | |
|---|---|---|---|---|---|---|---|---|---|
| | CP | EE | Ash | CP | EE | Ash | CF | NDF | ADF |
| **Drip irrigation levels (PE)** | | | | | | | | | |
| 0.6 | 14.50b | 2.83b | 6.79b | 5.87d | 1.18b | 2.79c | 24.65a | 36.68a | 29.48a |
| 0.8 | 15.78ab | 3.36ab | 9.29a | 7.56ab | 1.64a | 4.54ab | 20.52b | 31.48b | 23.63b |
| 1.0 | 17.18a | 3.51a | 9.86a | 8.85a | 1.81a | 5.09a | 19.56b | 30.05b | 22.18b |
| 0.6 + 0.8 | 15.19ab | 3.09ab | 8.64a | 6.60cd | 1.35ab | 3.74b | 22.93ab | 33.17ab | 25.97ab |
| 0.6 + 1.0 | 15.60ab | 3.34ab | 9.01a | 7.08bc | 1.52a | 4.16ab | 21.48ab | 32.10b | 24.57b |
| 0.8 + 1.0 | 16.72ab | 3.43ab | 9.46a | 8.45ab | 1.75a | 4.81a | 19.59b | 30.99b | 22.79b |
| S.Em.± | 0.71 | 0.29a | 0.54 | 0.47 | 0.16 | 0.33 | 1.11 | 1.22 | 1.28 |
| LSD$_{P=0.05}$ | 2.24 | ns | 1.69 | 1.47 | 0.52 | 1.03 | 3.49 | 3.84 | 4.03 |
| **Sulphur levels (kg ha$^{-1}$)** | | | | | | | | | |
| 0 | 15.21b | 3.01b | 7.37d | 6.74b | 1.35c | 3.32c | 23.36a | 34.66a | 27.01a |
| 20 | 15.71ab | 3.26ab | 8.86c | 7.32ab | 1.54b | 4.17b | 21.74b | 32.64a | 25.00bc |
| 40 | 16.13ab | 3.30a | 9.20b | 7.68a | 1.59a | 4.43ab | 20.51b | 31.53b | 23.89cd |
| 60 | 16.26a | 3.48a | 9.93a | 7.87a | 1.69a | 4.83a | 20.21b | 30.82b | 23.18d |
| S.Em.± | 0.33 | 0.06 | 0.22 | 0.22 | 0.04 | 0.14 | 0.44 | 0.68 | 0.57 |
| LSD$_{P=0.05}$ | 0.94 | 0.16 | 0.63 | 0.62 | 0.11 | 0.39 | 1.27 | 1.96 | 1.63 |
| **I x S $_{P=0.05}$** | 2.30 | 0.39 | 1.54 | 1.52 | 0.28 | 0.96 | 3.12 | 4.81 | 3.99 |

ha[-1] resulted in 3.3, 5.6, 7.1% (CP content), 8.3, 9.7, 15.6% (EE content) and 20.1, 24.9, 34.8% (ash content) higher in groundnut fodder, respectively, when computed over no S application. As per the interaction effect it was observed that drip irrigation level of 1.0 PE produced maximum CP, EE and ash content and yield when supplied with 60 kg S ha[-1], although it was at par with the 1.0 PE and 40 kg S ha[-1], 0.8+1.0 PE and 60 kg S ha[-1], 0.8+1.0 PE and 40 kg S ha[-1].

**Fiber content of groundnut fodder.** Results indicated that different levels of drip irrigation and sulphur fertilization significantly affected the content of NDF, ADF and CF (Tables 2 and 3). Lowest content of CF(19.92%), NDF(30.47%) and ADF(22.35%)was recorded in 1.0 PE level of drip irrigation, which was at par with 0.8 +1.0 PE level of drip irrigation with 19.93, 31.52 and 22.96% CF, NDF and ADF content, respectively, whereas, highest CF, NDF and ADF content was recorded in 0.6 PE level of irrigation. In general, Fiber contents were slightly higher in 2020 than 2021. Different levels of sulphur fertilization significantly influenced CF, NDF and ADF content in groundnut fodder during both the years (Tables 2 and 3). When compared with no fertilization control, application of higher doses of S (40 and 60 kg S ha[-1]) significantly decreased the Fiber content of crop during both the years of 2020 and 2021. The minimum CF, NDF and ADF content (20.64, 31.11 and 23.32%, respectively) was recorded with 60 kg S ha[-1] which was at par with 40 kg S ha[-1], whereas maximum CF content was recorded in control. On an average, application of 20, 40 and 60 kg S ha[-1] resulted in 7.3, 12.6, 14.4% lower CF content, 4.8, 9.7, 13.2% lower NDF content, 8.8, 13.7, 17.2% lower ADF content of groundnut, respectively, when computed over no S application. Fiber content was decreased with increasing levels of S from 0 to 60 kg S ha[-1], but the rate of decrease was higher from 0–20 kg S ha[-1], thereafter rate of decrease in Fiber content was proportionately lower. As per the interaction effect it was observed that drip irrigation level of 1.0 PE produced minimum CF, NDF and ADF content when supplied with 60 kg S ha[-1], although it was at par with the 1.0 PE and 40 kg S ha[-1], 0.8+1.0 PE and 60 kg S ha[-1], 0.8+1.0 PE and 40 kg S ha[-1].

**Different digestibility and quality indices.** All the digestibility and quality indices were significantly affected due to different levels of drip irrigation and sulphur fertilization during both the years (Tables 4 and 5). The groundnut fodder in 1.0 PE level of irrigation showed significantly higher DMI (3.98%), DMD (71.49%), TDN (72.50%) and $NE_L$ (7.18 MJ kg[-1]) content as compared to other remaining levels of drip irrigations, especially 0.6 PE level.

As per sulphur fertilization, application of 60 kg S ha[-1]reported significantly higher DMI (3.92%), DMD (70.73%), TDN (71.24%) and $NE_L$ (7.08 MJ kg[-1]) content followed by 40 kg S application ha[-1], whereas, lowest values were obtained in control. As per the interaction effect, 1.0 PE level of irrigation when supplied with 60 kg Sha[-1] resulted in maximum DMI, DMD, TDN and $NE_L$ contents, although it was at par with the 1.0 PE and 40 kg S ha[-1], 0.8+1.0 PE and 60 kg S ha[-1], 0.8+1.0 PE and 40 kg S ha[-1].

When the groundnut fodder crop grown under higher levels of drip irrigation i.e., 1.0 PE and 0.8 + 1.0 PE levels recorded higher RFV (221.5) (Fig 2), RFQ (236.0) (Fig 3), DE (13.93 MJ kg[-1]), ME (11.44MJ kg[-1]), DFE (13.35 MJ kg[-1]), and NE (6.03 MJ kg[-1]) as compared to other levels of drip irrigation (Tables 4 and 5). Among sulphur fertilization, groundnut fodder supplied with 60 kg S ha[-1]reported significantly higherRFV (215.8), RFQ (228.6), DE (13.80 MJ kg[-1]), ME (11.39MJ kg[-1]), DFE (13.11 MJ kg[-1]), and NE (5.86 MJ kg[-1]). As per the interaction effect, 1.0 PE level of irrigation when supplied with 60 kg Sha[-1] resulted in maximum RFV, RFQ, DE, ME, DFE and NE contents, although it was at par with the 1.0 PE and 40 kg S ha[-1], 0.8+1.0 PE and 60 kg S ha[-1], 0.8+1.0 PE and 40 kg S ha[-1].

**Nutrient content and uptake in groundnut fodder.** Nitrogen and sulphur content was estimated in dry fodder and it remained unaffected due to drip irrigation levels during both the years (Table 6). On an average, it can be seen that N and S content in dry fodder was higher in 1.0 PE drip irrigation, which was statistically at par with 0.8 +1.0 PE level of drip irrigation,

**Table 4. Effect of drip irrigation and sulphur levels on digestibility indices of groundnut fodder during 2020; DMI-dry matter intake, DMD- dry matter digestibility, TDN- total digestible nutrients, NEL- net energy for lactation, DE- digestible energy, ME- metabolizable energy, DFE- digestible feed energy, NE- net energy.**

| | DMI (%) | DMD (%) | TDN (%) | NE$_L$ (Mcalkg$^{-1}$) | NE$_L$ (MJ kg$^{-1}$) | DE (Mcalkg$^{-1}$) | DE (MJ kg$^{-1}$) | ME (MJ kg$^{-1}$) | DFE (MJ kg$^{-1}$) | NE ((MJ kg$^{-1}$) |
|---|---|---|---|---|---|---|---|---|---|---|
| **Drip irrigation levels (PE)** | | | | | | | | | | |
| 0.6 | 3.15b | 65.67b | 62.85b | 1.52b | 6.36b | 3.08b | 12.89b | 10.58b | 11.57b | 4.74b |
| 0.8 | 3.71a | 70.22a | 70.40a | 1.67a | 7.00a | 3.28a | 13.70a | 11.25a | 12.96a | 5.75a |
| 1.0 | 3.92a | 71.36a | 72.28a | 1.71a | 7.16a | 3.32a | 13.91a | 11.42a | 13.31a | 6.00a |
| 0.6 + 0.8 | 3.53ab | 68.41ab | 67.39ab | 1.61ab | 6.74ab | 3.20ab | 13.38ab | 10.98ab | 12.41ab | 5.35ab |
| 0.6 + 1.0 | 3.66a | 69.51a | 69.21a | 1.65a | 6.90a | 3.25a | 13.58a | 11.15a | 12.74a | 5.59a |
| 0.8 + 1.0 | 3.79a | 70.89a | 71.50a | 1.70a | 7.09a | 3.30a | 13.82a | 11.35a | 13.16a | 5.90a |
| S.Em.± | 0.14 | 1.00 | 1.66 | 0.03 | 0.14 | 0.04 | 0.18 | 0.15 | 0.31 | 0.22 |
| LSD$_{P=0.05}$ | 0.43 | 3.15 | 5.22 | 0.11 | 0.44 | 0.13 | 0.56 | 0.46 | 0.96 | 0.70 |
| **Sulphur levels (kg ha$^{-1}$)** | | | | | | | | | | |
| 0 | 3.39b | 67.44c | 65.79c | 1.58c | 6.61c | 3.16c | 13.21c | 10.84c | 12.11c | 5.14c |
| 20 | 3.50b | 69.22b | 68.73b | 1.64b | 6.86b | 3.23b | 13.52b | 11.10b | 12.65b | 5.53b |
| 40 | 3.72a | 70.08ab | 70.17a | 1.67ab | 6.98ab | 3.27ab | 13.68ab | 11.23ab | 12.92ab | 5.72ab |
| 60 | 3.89a | 70.62a | 71.06a | 1.69a | 7.06a | 3.29a | 13.78a | 11.31a | 13.08a | 5.84a |
| S.Em.± | 0.06 | 0.43 | 0.72 | 0.01 | 0.06 | 0.02 | 0.08 | 0.06 | 0.13 | 0.10 |
| LSD$_{P=0.05}$ | 0.18 | 1.25 | 2.06 | 0.04 | 0.18 | 0.05 | 0.22 | 0.18 | 0.38 | 0.28 |
| **I x S $_{P=0.05}$** | 0.43 | 3.05 | 5.07 | 0.10 | 0.43 | 0.13 | 0.54 | 0.45 | 0.93 | 0.67 |

whereas lowest N and S content was recorded in 0.6 PE level of irrigation. Application of higher doses of S (40 and 60 kg S ha$^{-1}$) contributed positively in better vegetative, reproductive and root growth of the crop resulting in higher N and S content in groundnut fodder during both the years of 2020 and 2021. The maximum N (1.48%) and S (0.24%) content was recorded with 60 kg S ha$^{-1}$ which was at par with 40 kg S ha$^{-1}$ (1.46% and0.23%), whereas lowest N and S

**Table 5. Effect of drip irrigation and sulphur levels on digestibility indices of groundnut fodder during 2021; DMI-dry matter intake, DMD- dry matter digestibility, TDN- total digestible nutrients, NEL- net energy for lactation, DE- digestible energy, ME- metabolizable energy, DFE- digestible feed energy, NE- net energy.**

| | DMI (%) | DMD (%) | TDN (%) | NE$_L$ (Mcalkg$^{-1}$) | NE$_L$ (MJkg$^{-1}$) | DE (Mcalkg$^{-1}$) | DE (MJkg$^{-1}$) | ME (MJ kg$^{-1}$) | DFE (MJ kg$^{-1}$) | NE ((MJ kg$^{-1}$) |
|---|---|---|---|---|---|---|---|---|---|---|
| **Drip irrigation levels (PE)** | | | | | | | | | | |
| 0.6 | 3.30b | 65.93b | 63.29b | 1.53b | 6.39b | 3.09b | 12.94b | 10.62b | 11.65b | 4.80b |
| 0.8 | 3.84a | 70.50a | 70.85a | 1.68a | 7.04a | 3.29a | 13.75a | 11.29a | 13.04a | 5.81a |
| 1.0 | 4.05a | 71.62a | 72.71a | 1.72a | 7.20a | 3.34a | 13.95a | 11.46a | 13.39a | 6.06a |
| 0.6 + 0.8 | 3.65ab | 68.67ab | 67.83ab | 1.62ab | 6.78ab | 3.21ab | 13.43ab | 11.02ab | 12.49ab | 5.41ab |
| 0.6 + 1.0 | 3.77ab | 69.76a | 69.63a | 1.66a | 6.93a | 3.26a | 13.62a | 11.18a | 12.82a | 5.65a |
| 0.8 + 1.0 | 3.91a | 71.15a | 71.93a | 1.70a | 7.13a | 3.32a | 13.87a | 11.39a | 13.24a | 5.96a |
| S.Em.± | 0.17 | 1.00 | 1.65 | 0.03 | 0.14 | 0.04 | 0.18 | 0.15 | 0.30 | 0.22 |
| LSD$_{P=0.05}$ | 0.52 | 3.14 | 5.20 | 0.11 | 0.44 | 0.13 | 0.56 | 0.46 | 0.96 | 0.70 |
| **Sulphur levels (kg ha$^{-1}$)** | | | | | | | | | | |
| 0 | 3.51c | 67.86c | 66.48c | 1.59b | 6.67c | 3.17c | 13.28c | 10.90b | 12.24c | 5.23c |
| 20 | 3.71bc | 69.43b | 69.08b | 1.65a | 6.89b | 3.24b | 13.56b | 11.13a | 12.72b | 5.57b |
| 40 | 3.85ab | 70.29ab | 70.51ab | 1.68a | 7.01ab | 3.28ab | 13.72ab | 11.26a | 12.98ab | 5.77ab |
| 60 | 3.95a | 70.84a | 71.43a | 1.69a | 7.09a | 3.30a | 13.82a | 11.34a | 13.15a | 5.89a |
| S.Em.± | 0.08 | 0.44 | 0.73 | 0.01 | 0.06 | 0.02 | 0.08 | 0.07 | 0.14 | 0.10 |
| LSD$_{P=0.05}$ | 0.23 | 1.27 | 2.10 | 0.04 | 0.18 | 0.05 | 0.23 | 0.19 | 0.39 | 0.28 |
| **I x S $_{P=0.05}$** | 0.57 | 3.10 | 5.15 | 0.10 | 0.44 | 0.13 | 0.57 | 0.46 | 0.95 | 0.69 |

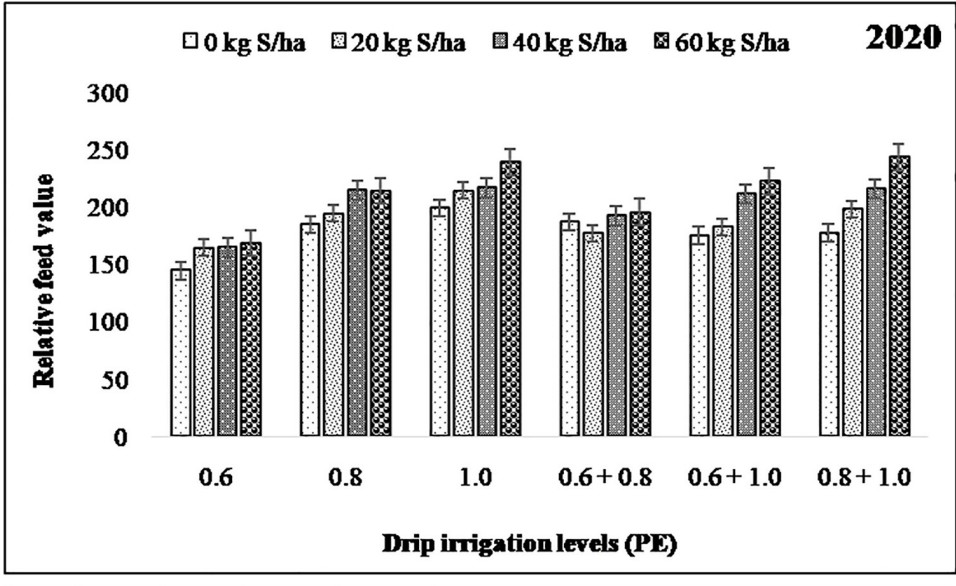

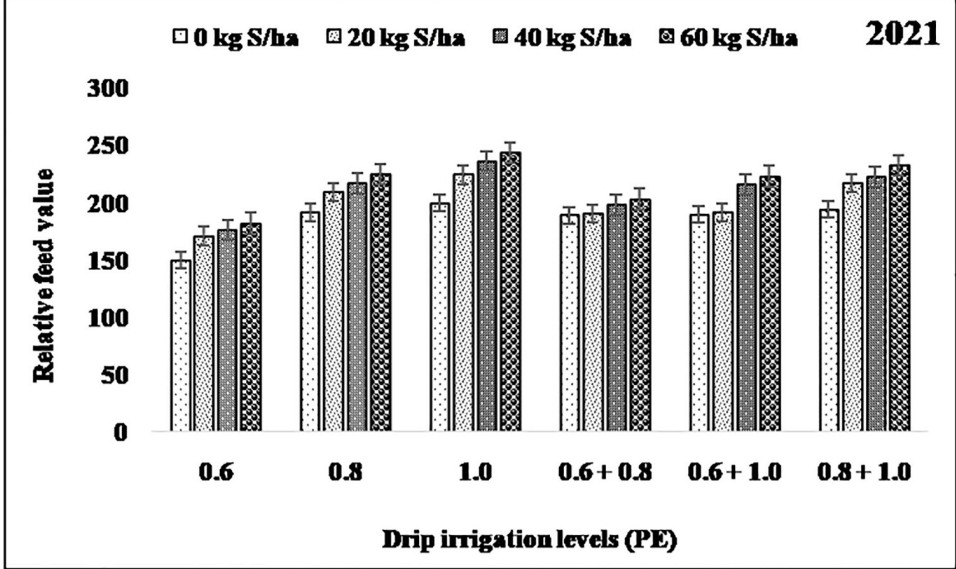

**Fig 2. Relative feed value of groundnut dry fodder as influenced by levels of drip irrigation and sulphur fertilization.**

content was recorded in control. Nitrogen and sulphur content increased with increasing levels of S from 0 to 60 kg S ha[-1], but the rate of increase was higher from 0–20 kg S ha[-1], thereafter rate of increase in N and S content was proportionately lower following the Baule unit concept.

Drip irrigation and S fertilization levels had significant effect on N and S uptake of groundnut fodder during both the years (Table 6)because nutrient uptake is governed by fodder yield of groundnut and yield was significantly differed due to treatment effect, so as N and S uptake. Results indicated that Nand S uptake by fodder was significantly higher in 1.0 PE drip irrigation (77.8 and 11.75 kg ha[-1], respectively), which was statistically at par with 0.8 +1.0 PE level of drip irrigation (75.1 and 11.66 kg ha[-1], respectively), whereas lowest N and S uptake was

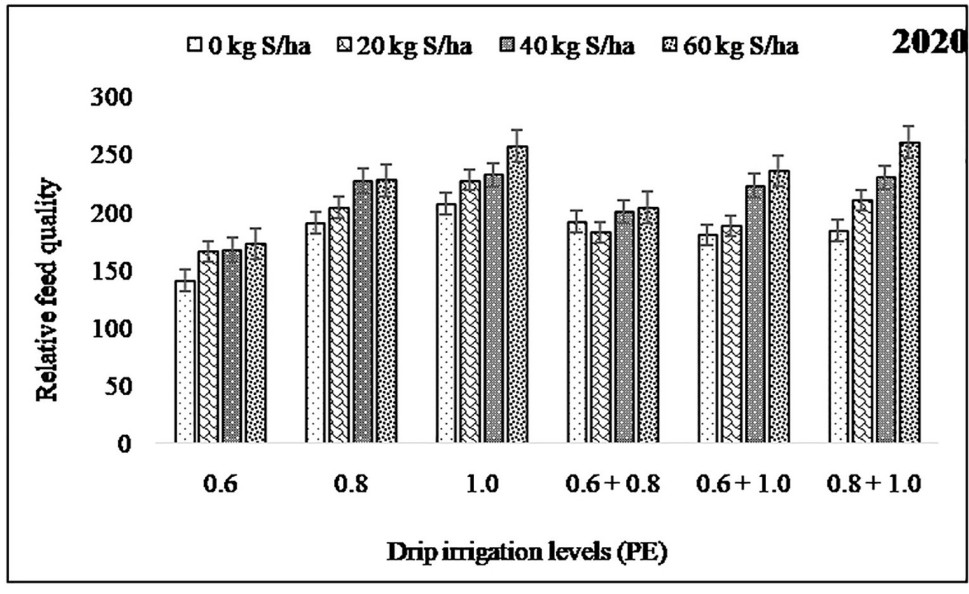

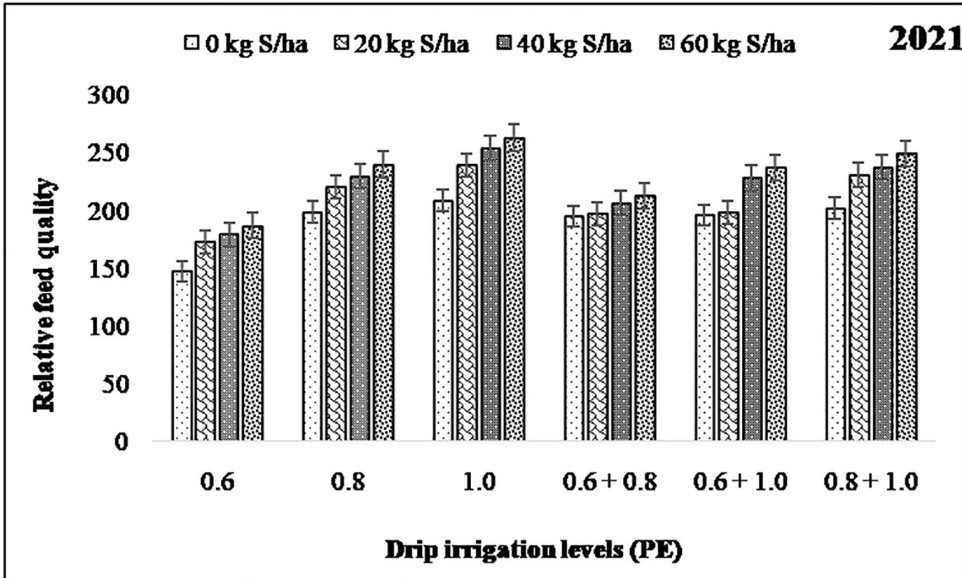

**Fig 3. Relative feed quality of groundnut dry fodder as influenced by levels of drip irrigation and sulphur fertilization.**

recorded in 0.6 PE level of irrigation (38.7 and 8.91 kg ha$^{-1}$, respectively). In general, total N uptake was slightly higher in 2021 than 2020, because of higher yield in 2021. Application of higher doses of S (40 and 60 kg S ha$^{-1}$) resulted in higher N and S uptake by groundnut fodder during both the years. The maximum N and S uptake (70.5 and 11.77 kg ha$^{-1}$, respectively) was recorded with 60 kg S ha$^{-1}$ which was at par with 40 kg S ha$^{-1}$ (68.4 and 11.11 kg ha$^{-1}$), whereas lowest N and S uptake (60.4 and 8.91 kg ha$^{-1}$) was recorded in control. As per the interaction effect it was observed that drip irrigation level of 1.0 PE when supplied with 60 kg S ha$^{-1}$ resulted in maximum and N and S uptake, although it was at par with the 1.0 PE and 40 kg S ha$^{-1}$, 0.8+1.0 PE and 60 kg S ha$^{-1}$, 0.8+1.0 PE and 40 kg S ha$^{-1}$.

**Table 6. Effect of drip irrigation and sulphur levels on nutrient content and uptake of groundnut fodder during 2020 and 2021.**

| | N content | | S content | | N uptake | | S uptake | |
|---|---|---|---|---|---|---|---|---|
| | 2020 | 2021 | 2020 | 2021 | 2020 | 2021 | 2020 | 2021 |
| **Drip irrigation levels (PE)** | | | | | | | | |
| 0.6 | 1.336 | 1.350 | 0.216 | 0.218 | 51.1c | 54.2c | 8.63b | 9.19b |
| 0.8 | 1.442 | 1.430 | 0.231 | 0.227 | 65.9abc | 68.4abc | 10.89ab | 11.27ab |
| 1.0 | 1.547 | 1.535 | 0.234 | 0.231 | 76.6a | 79.0a | 11.57a | 11.93a |
| 0.6 + 0.8 | 1.403 | 1.370 | 0.224 | 0.223 | 60.5bc | 59.2b | 9.65ab | 9.70ab |
| 0.6 + 1.0 | 1.431 | 1.411 | 0.230 | 0.229 | 64.4abc | 63.9abc | 10.36ab | 10.48ab |
| 0.8 + 1.0 | 1.521 | 1.502 | 0.233 | 0.231 | 74.1ab | 76.1a | 11.49a | 11.83ab |
| S.Em.± | 0.094 | 0.084 | 0.039 | 0.039 | 4.8 | 4.9 | 0.86 | 2.06 |
| CD$_{P=0.05}$ | ns | ns | ns | ns | 15.0 | 15.2 | 2.64 | 2.72 |
| **Sulphur levels (kg ha$^{-1}$)** | | | | | | | | |
| 0 | 1.394 | 1.386 | 0.203 | 0.202 | 59.5b | 61.3c | 8.76c | 9.05c |
| 20 | 1.435 | 1.416 | 0.227 | 0.225 | 64.5ab | 65.5bc | 10.38b | 10.69b |
| 40 | 1.469 | 1.450 | 0.236 | 0.235 | 68.0a | 68.9ab | 10.95a | 11.27a |
| 60 | 1.487 | 1.479 | 0.246 | 0.244 | 69.6a | 71.4a | 11.63a | 11.92a |
| S.Em.± | 0.023 | 0.025 | 0.003 | 0.004 | 1.9 | 1.8 | 0.25 | 0.26 |
| CD$_{P=0.05}$ | ns | ns | ns | ns | 5.3 | 5.2 | 0.71 | 0.75 |

**Water use efficiency.** Water use efficiency was computed by dividing green and dry fodder yield with their respective water use of the treatment. It was found that lower the water use, higher the WUE of the treatment. Data presented in Table 7 revealed that different treatments of drip irrigation levels and S fertilization significantly influenced water use efficiency of

**Table 7. Effect of drip irrigation and sulphur levels on water use efficiency and uptake of groundnut fodder during 2020 and 2021.**

| Drip irrigation levels (PE) | Sulphur levels (kg ha$^{-1}$) | | | | | | | |
|---|---|---|---|---|---|---|---|---|
| | 0 | 20 | 40 | 60 | 0 | 20 | 40 | 60 |
| | 2020 | | | | 2021 | | | |
| **Water use efficiency$_{GFY}$(kg ha$^{-1}$ mm)** | | | | | | | | |
| **0.6** | 14.47de | 15.29bcde | 17.66ab | 17.91a | 15.49de | 16.33bcde | 18.42ab | 18.96a |
| **0.8** | 15.28bcde | 15.35abcde | 14.74cde | 16.75abcd | 16.08bcde | 16.08abcde | 15.47cde | 17.49abcd |
| **1.0** | 13.76e | 14.36de | 14.91cde | 14.69de | 14.38e | 14.98de | 15.53cde | 15.33de |
| **0.6 + 0.8** | 15.03cde | 17.31abc | 17.69a | 15.29bcde | 15.13cde | 17.41abc | 17.79a | 15.39bcde |
| **0.6 + 1.0** | 14.01e | 14.87cde | 15.32abcde | 16.18abcde | 14.14e | 14.99cde | 15.42abcde | 16.17abcde |
| **0.8 + 1.0** | 14.67de | 15.43abcde | 15.12bcde | 15.45abcde | 15.28de | 16.05abcde | 15.73bcde | 16.06abcde |
| **S.Em.±** | 0.90 | | | | 0.86 | | | |
| **CD$_{P=0.05}$** | 2.59 | | | | 2.46 | | | |
| **Water use efficiency$_{DFY}$(kg ha$^{-1}$ mm)** | | | | | | | | |
| **0.6** | 4.64def | 4.90bcde | 5.66ab | 5.74a | 4.93def | 5.20bcde | 5.87ab | 6.04a |
| **0.8** | 4.72cdef | 4.75cdef | 4.56cdef | 5.18abc | 4.97cdef | 4.97cdef | 4.78cdef | 5.40abc |
| **1.0** | 3.98f | 4.16ef | 4.31ef | 4.25ef | 4.16f | 4.33ef | 4.49ef | 4.43ef |
| **0.6 + 0.8** | 4.49def | 5.17abcd | 5.29abc | 4.57cdef | 4.52def | 5.20abcd | 5.31abc | 4.59cdef |
| **0.6 + 1.0** | 4.15ef | 4.41def | 4.54cdef | 4.80cde | 4.19ef | 4.44def | 4.56cdef | 4.79cde |
| **0.8 + 1.0** | 4.29ef | 4.52cdef | 4.43def | 4.52cdef | 4.47ef | 4.69cdef | 4.60def | 4.70cdef |
| **S.Em.±** | 0.27 | | | | 0.26 | | | |
| **CD$_{P=0.05}$** | 0.79 | | | | 0.75 | | | |

groundnut fodder during both the years. Significantly highest WUE was recorded in 0.6 PE drip irrigation level and 60 kg S ha$^{-1}$ (18.44 and 5.89 kg ha$^{-1}$mm with GFY and DFY, respectively) during both the years. However, 0.6 PE level of irrigation was at par with 0.8 PE and 0.6 +1.0 PE level of drip irrigation, irrespective of the year and fodder yield. Similarly, WUE in 60 kg S ha$^{-1}$ was at par with 40 kg S ha$^{-1}$during both the years. Lowest WUE was recorded in 1.0 PE level of irrigation and when no sulphur was applied.

## Discussion

### Effect of drip irrigation and sulphur on fodder yield and nutrient uptake

Shortage of water is the major constraint for limiting crop yield in arid and semi-arid areas [31]and improving effective utilization of water is an urgent need for sustainable crop production in these areas [32]. Deficit irrigation (DI) has been emerging as an effective practice to improve water use efficiency, and saving of water [9, 33]. Priorto this study, little information exists on fodder productivity, water use efficiency, and quality of fodder and cake of groundnut under varying irrigation and S application rates. Results of the present study revealed that S application is effective in increasing yield of groundnut in the areas having low content of S in soil, and to achieve optimal coupling effect of irrigation and S for optimizing yield, aproper combination of irrigation and fertilizer is required. Decrease in soil moisture often makes it difficult for uptake of nutrients such as N, P, K and S, reducing growth, development and yield. Lower photosynthesis and cell growth under moisture scarce conditions lowers the growth and yield of crops grown [34].

Minimum fodder yield was recorded in 0.6 PE level of drip irrigation, with the increasing PE levels fodder yield was increased, recording significantly higher yield in 1.0 PE drip irrigation followed by 0.8 +1.0 PE level of drip irrigation. Dry matter production and its partitioning to sinks is the major determining factor of yield; and water-deficit condition largely affects them. Similar observation was recorded in the study that yield declined with a reduction in irrigation regime (0.6 PE level). Higher irrigation levels of 1.0 PE and 0.8+1.0 PE helped in maintaining the stress-free conditions for optimum growth and development of plants throughout the crop growing period. The better response may be due to more frequent and optimum supply of irrigation water, which not only increased nutrients availability in root zone but also enhanced uptake of nutrients by the plant as well as ensured better partitioning of nutrients in actively growing plant parts resulting in better growth, biomass and yield contributing characters. The yield increase with higher PE levels of irrigation is due to frequent higher volume of water application through drip irrigation which resulted in favour able microclimate and kept soil moisture constantly nearer to field capacity which helped in increasing yields. Proper irrigation scheduling under drip provides means of reducing water wastage through evaporation with increased yields [35] as of treatment 0.8+1.0PE in the present study. The maintenance of continuously high soil water potential, thus minimizing wide fluctuations in soil water content might be the reason of yield increase. Nitrogen and sulphur content and uptake in haulm or dry fodder was significantly higher in 1.0 PE drip irrigation, which was statistically at par with 0.8 +1.0 PE level of drip irrigation, whereas lowest uptake was recorded in 0.6 PE level of irrigation. Higher vegetative growth of shoots and roots resulted in the better nutrient uptake under higher irrigation regimes. The nutrient uptake is a function of nutrient concentration in economic and biological parts of the crop. The increase in N and S uptake by crop might have ascribed to the cumulative effect for enhanced nutrient concentration and biomass yield [36].

The sulphur fertiliser has a positive impact on the fodder yield of groundnut with higher uptake of other micro and macronutrients [21]. Sulphur nutrition to crop is vital both from a

quality and quantity point of view. Sulphur lowers the HCN content of certain crops, promotes nodulation in legumes, and increases fodder yield in oilseeds like groundnut. Higher fodder yield with increased application of sulphur also attributed to protein and enzyme synthesis as it is a constituent of sulphur containing amino acids namely methionine, cysteine, and cystine [37]. According to Yadav et al. [38] with increasing levels of S, fodder yield and quality of groundnut was increased due to better functioning of the roots and improving the sulphur uptake in the root zone.

## Effect of drip irrigation and sulphur on fodder quality parameters

For better quality and palatability of fodder, crude protein content, ash content and ether extract should be higher, whereas, all the Fiber, CF, NDF, ADF and lignin content should be lower in fodder. Fibers has the inverse relationship with crude protein and ash i.e. higher crude Fiber, lower the crude protein in fodder, so is the poor quality and palatability of fodder. It was evident from data all the dry fodder quality parameters were improved under higher irrigation regimes and higher doses of S application during both the years. According to [39, 40] forage containing lower concentrations of ADF and NDF and higher DMI are of good quality and NDF and ADF are positively correlated with shorter irrigation intervals that means reduction in irrigation regime or interval will increase NDF and ADF content. Similar finding was also reported that NDF and ADF in alfalfa was increased by reducing irrigation [41]. Fiber content in plants is affected by growth stage, leaf to stem ratio, nutrient availability in soil and prevalent weather [42]. The increase in crude Fiber in lower level of irrigation may be associated with reduction in photosynthates [43]. The NDF and ADF concentration was lower in higher level of irrigation regimes with 40 kg S ha$^{-1}$ could be explained by the higher concentration of carbohydrate leading to better vegetative growth [44]. Under water stress, a main physiological response of plant is to increase the insoluble Fibers in cell walls to prevent moisture loss. Providing regular and continuous irrigation water as in drip irrigation slows down this process and prevents increase in crude Fiber [45]. The percentage of ash in forage represents the amount of minerals in plant tissues with the absorption of these materials by roots generally diminishing under drought conditions [46]. With the irrigation regime 0.8 +1.0 PE and 40 kg S ha$^{-1}$ increased the ash content, due to optimum supply and availability of nutrients, more nutrient uptake, by changing the pHor secretion of enzymes [47].

The optimum application of Scan be able to sufficient supply of nutrients for protein synthesis [48] thus augmenting the crude protein in groundnut fodder. Numerous studies have reported that when drought stress increases, the CP content of forage improves due to accumulation of nitrogen [49]. The significantly highest crude protein, ash content and ether extract and lowest crude Fiber, NDF, ADF and lignin content was recorded with application of higher doses of S, because application of S contributed positively in better vegetative, reproductive and root growth of the crop resulting in higher nutrient content in haulm leading to better quality of dry fodder. Therefore, S and N content was positively correlated with CP and ash content, however, negatively correlated with CF, NDF, ADF and lignin (Figs 4 and 5), indicating that improvement in S fertilization, significantly improved the quality of groundnut haulm/dry fodder.

The groundnut fodder yield and quality were increased when 0.8+1.0 PE level of irrigation was applied with 40 kg S ha$^{-1}$, which may increase the leaf water potential, the rate of $CO_2$ assimilation, transpiration, cause root growth and enhanced water absorption in the plant. Thus, it can be stated that combined application of irrigation and S would probably improve the absorption of nutrients and water in the plant thereby boosting its growth, development and increasing the green and dry fodder yield of groundnut in arid environment.

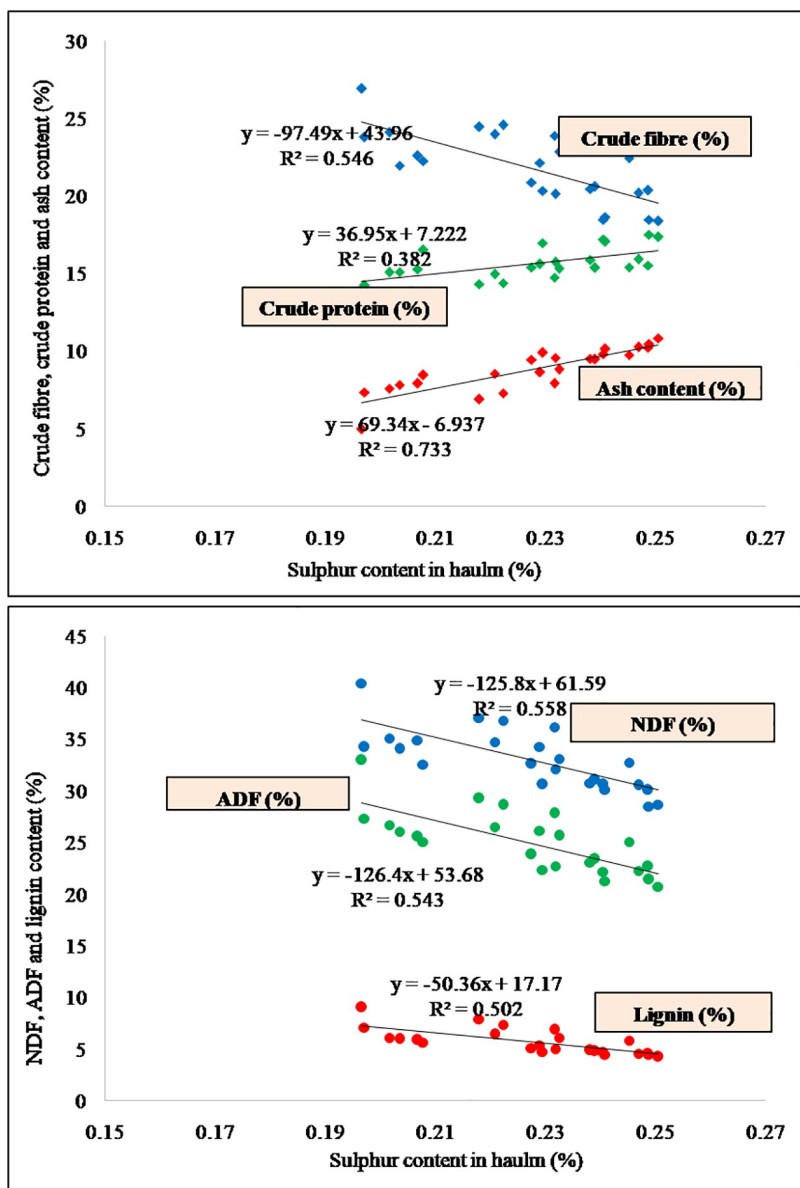

**Fig 4. Relationship of sulphur content in haulm/dry fodder with fodder quality parameters.**

## Conclusions

It can be concluded that lower irrigation regimes (0.6 PE) reduced the yield of fodder and increased the Fiber content and decreased the protein and ash content leading to poor quality fodder with low digestibility. Adjustment in irrigation regime i.e. 0.8PE (from sowing to 45 DAS) + 1.0 PE (46 DAS to maturity) resulted in better productivity, water saving and nutritious fodder. Crop responded to increasing levels of S from 0–60 kg ha[-1] but response was higher from 0–40 kg S ha[-1], thereafter response was proportionately lower. Coupling effect of 0.8 +1.0 PE and 40 kg S ha[-1] improved fodder yield and quality due to optimum supply and availability of moisture and nutrients leading to lower Fiber content with better digestibility of the fodder.

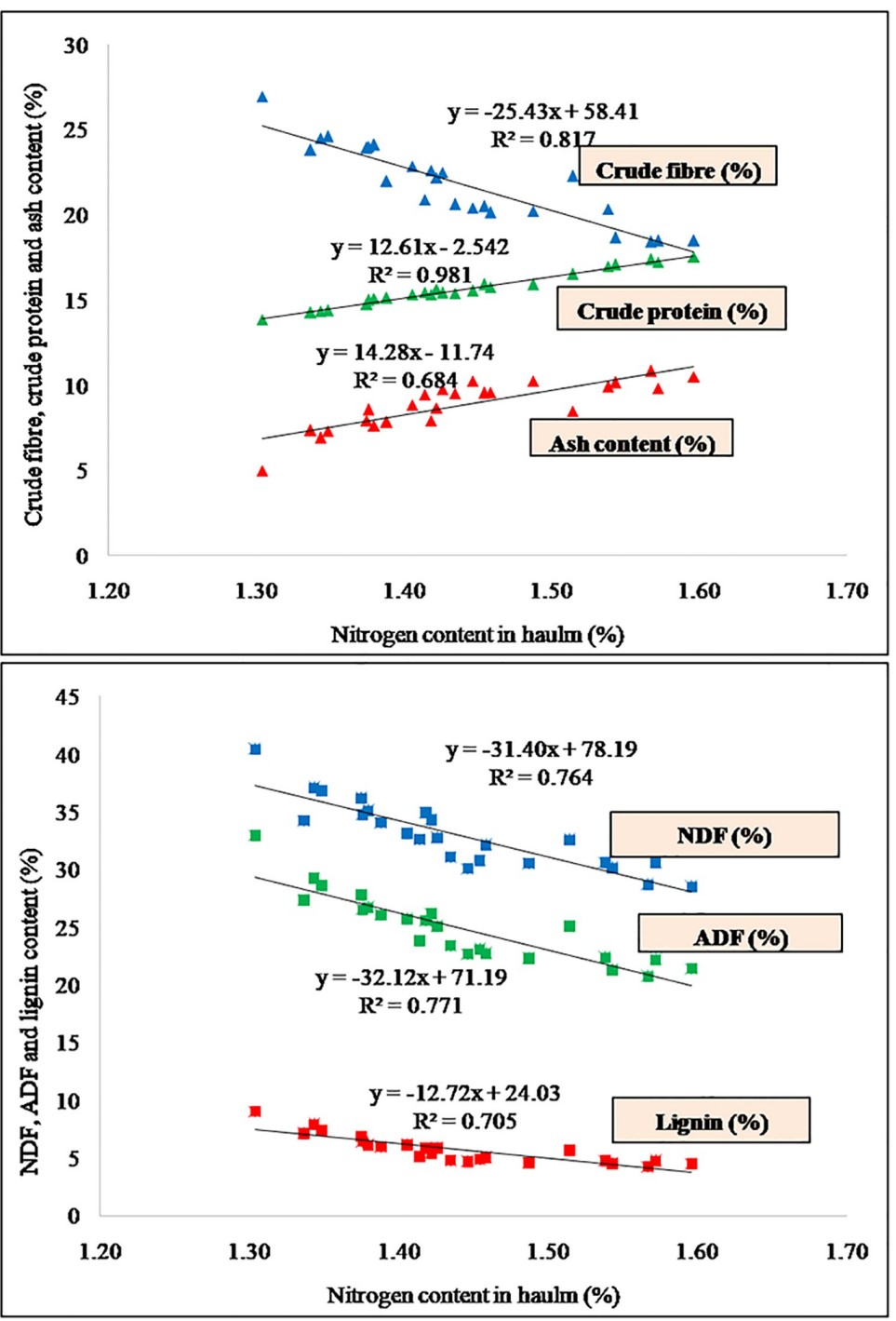

**Fig 5. Relationship of nitrogen content in haulm/dry fodder with fodder quality parameters.**

## Acknowledgments

The authors want to acknowledge Dean, College of Agriculture, Bikaner and Vice-chancellor, SKRAU, Bikaner for providing necessary facilities and support for completion of the study. I also want to thank Director, NRCC, Bikaner for granting study leave for this investigation.

## Author Contributions

**Conceptualization:** Priyanka Gautam, S. R. Bhunia.

**Data curation:** Priyanka Gautam, S. R. Bhunia, B. Lal.

**Formal analysis:** R. K. Sawal.

**Investigation:** Priyanka Gautam, V. K. Yadav, Gograj.

**Methodology:** V. K. Yadav, Ramniwas, Gograj.

**Project administration:** Priyanka Gautam, A. Sahoo.

**Resources:** S. R. Bhunia.

**Software:** Shantanu Rakshit, Rajesh Bishnoi.

**Supervision:** A. Sahoo, V. S. Rathore.

**Validation:** Priyanka Gautam.

**Visualization:** R. K. Sawal, Shantanu Rakshit, Ramniwas, V. S. Rathore.

**Writing – original draft:** Priyanka Gautam, B. Lal.

**Writing – review & editing:** Priyanka Gautam, A. Sahoo, R. K. Sawal, Shantanu Rakshit, B. Lal, Rajesh Bishnoi, V. S. Rathore.

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
