## [Decision Letter · Decision Letter 0]

8 Mar 2023

PONE-D-23-01558Drip irrigation and sulphur fertilization influenced fodder yield, quality and water use efficiency of groundnut in arid regionPLOS ONE

Dear Dr. Gautam,

Thank you for submitting your manuscript to PLOS ONE. After careful consideration, we feel that it has merit but does not fully meet PLOS ONE’s publication criteria as it currently stands. Therefore, we invite you to submit a revised version of the manuscript that addresses the points raised during the review process.

We look forward to receiving your revised manuscript.

Kind regards,

Arun Kumar Shanker

Academic Editor

PLOS ONE

Journal Requirements:

Additional Editor Comments:

The reviewers have suggested revision, the authors are requested to incorporate the suggestions and submit the revised MS

Reviewers' comments:

Reviewer's Responses to Questions

**Comments to the Author**

1. Is the manuscript technically sound, and do the data support the conclusions?

Reviewer #1: Yes

Reviewer #2: Yes

Reviewer #3: Yes

2. Has the statistical analysis been performed appropriately and rigorously? 

Reviewer #1: Yes

Reviewer #2: Yes

Reviewer #3: I Don't Know

3. Have the authors made all data underlying the findings in their manuscript fully available?

Reviewer #1: Yes

Reviewer #2: Yes

Reviewer #3: Yes

4. Is the manuscript presented in an intelligible fashion and written in standard English?

Reviewer #1: Yes

Reviewer #2: No

Reviewer #3: Yes

5. Review Comments to the Author

Reviewer #1: Your paper "Drip irrigation and sulphur fertilization influenced fodder yield, quality and water use efficiency of groundnut in arid region" addresses the fodder quality of groundnut. Problem of fodder deficiency and quality is severe in arid regions. Paper is written well. Introduction describes a clear background and significance of the problem to be solved. Methodology is described thoroughly. Results and discussion section written well. However, it needs to be revised to check spell mistakes and for language improvement. In conclusion, only a few lines are enough describing key findings of the study.

Reviewer #2: Please address the following suggestions and comments

Line 14: “However, no researchershavegiven emphasis” insert space between words.

Line 22: Define PE in Abstract

Line 58: “20-22 irrigation” Author means no. of irrigations? Too high? Cite it if yes, please!

Line 62: “regions of India.Drip irrigation” insert space after full stop.

Line 115: Define FIR ?

Line 185: Typo error in unit of WUE.

Line 202: Table 1, Add letters to show significant differences between treatment means.

Line 233: Table 2, Add letters to show significant differences between treatment means.

Line 258: Table 3, Add letters to show significant differences between treatment means.

Line 273: Table 4, Add letters to show significant differences between treatment means.

Line 275: Table 5, Add letters to show significant differences between treatment means.

Line 305: Table 6, Add letters to show significant differences between treatment means.

Line 332: Table 7, Add letters to show significant differences between treatment means.

Line 336: “Effect drip irrigation” Effect of drip irrigation

Line 379: “Effect drip irrigation” Effect of drip irrigation

Line 426: “0-60 kg ha-1” 0-60 kg ha-1

Reviewer #3: The research investigated the effect of drip irrigation and sulphur fertilization on fodder yield, quality and water use efficiency of groundnut in arid region. It tested the hypothesis that these two factors, at suitable levels, can save enough water and improve forage quality of groundnut in hot arid region where water and forage are important resources.

I consider the research relevant and suitable for publication in PLOS ONE. However, I think the manuscript can be improved further. I therefore have the following suggestions.

1. The manuscript needs to be further proof read to improve the English further. Few grammatical errors or omissions such as in line 1, 38, 49, 92-93, 194, 222-223, 352, 369 etc can be avoided.

2. Please include the reference to the point raised in line 77-79, ‘In most of the 78 groundnut growing tracts, the level of available sulphur reaches below the critical limit and groundnut crop is bound to suffer on account of sulphur deficiency.’

3. In line 95-97, the data presented was minimum to maximum rather than maximum to minimum e.g ‘The maximum and minimum temperature ranged between 26.6℃ to 43.2℃ and 8.5℃ to 31.1℃

4. It will be nice if the authors present the initial soil test of the location used. Authors can provide them if available

5. In your statistical analysis, please confirm that the data followed normal distribution and equal variances, which justifies the use of ANOVA.

5. In your results and tables, I will prefer you include the exact P values for the main effects and the interactions. This will make comparism easier.

6. When reporting the specifics in the first few tables, you did not mention which planting year you are referring to. One can only generally assume that both years followed the same trend, but this was also not mentioned by the authors.

7. Thank you.

6. PLOS authors have the option to publish the peer review history of their article (what does this mean?). If published, this will include your full peer review and any attached files.

Reviewer #1: No

Reviewer #2: No

Reviewer #3: No

---

## [Author Response · Author response to Decision Letter 0]

11 Apr 2023

Reviewer 1

Your paper "Drip irrigation and sulphur fertilization influenced fodder yield, quality and water use efficiency of groundnut in arid region" addresses the fodder quality of groundnut. Problem of fodder deficiency and quality is severe in arid regions. Paper is written well. Introduction describes a clear background and significance of the problem to be solved. Methodology is described thoroughly. Results and discussion section written well. 

However, it needs to be revised to check spell mistakes and for language improvement. In conclusion, only a few lines are enough describing key findings of the study. 

We want to thank reviewers for appreciating our reports. Agreed and complied; Manuscript is revised for language and spellings, extra lines were deleted from conclusion.

Reviewer 2

Line 14: “However, no researchershavegiven emphasis” insert space between words. 

Agreed and complied

Line 22: Define PE in Abstract. 

Agreed and complied

Line 58: “20-22 irrigation” Author means no. of irrigations? Too high? Cite it if yes, please! 

corrected the line and reference is provided for rest sentence

Line 62: “regions of India.Drip irrigation” insert space after full stop 

Agreed and complied

Line 115: Define FIR?

It is a seed treatment sequence of fungicide, insecticide, Rhizobium, Defined in text also

Line 185: Typo error in unit of WUE. 

Agreed and complied, unit is changed accordingly

Line 202: Table 1, Add letters to show significant differences between treatment means. 

Agreed and complied

Line 233: Table 2, Add letters to show significant differences between treatment means. 

Agreed and complied

Line 258: Table 3, Add letters to show significant differences between treatment means. 

Agreed and complied

Line 273: Table 4, Add letters to show significant differences between treatment means. 

Agreed and complied

Line 275: Table 5, Add letters to show significant differences between treatment means. 

Agreed and complied

Line 305: Table 6, Add letters to show significant differences between treatment means. 

Agreed and complied

Line 332: Table 7, Add letters to show significant differences between treatment means. 

Agreed and complied

Line 336: “Effect drip irrigation” Effect of drip irrigation 

Agreed and complied

Line 379: “Effect drip irrigation” Effect of drip irrigation 

Agreed and complied

Line 426: “0-60 kg ha-1” 0-60 kg ha-1 

Agreed and complied

Reviewer 3

 The manuscript needs to be further proof read to improve the English further. Few grammatical errors or omissions such as in line 1, 38, 49, 92-93, 194, 222-223, 352, 369 etc can be avoided. 

Agreed and complied; Manuscript is revised for language and spellings for improving the English.

Please include the reference to the point raised in line 77-79, ‘In most of the 78 groundnut growing tracts, the level of available sulphur reaches below the critical limit and groundnut crop is bound to suffer on account of sulphur deficiency. Agreed and complied; Reference is added 

In line 95-97, the data presented was minimum to maximum rather than maximum to minimum e.g ‘The maximum and minimum temperature ranged between 26.6℃ to 43.2℃ and 8.5℃ to 31.1℃ 

Agreed and complied;

It will be nice if the authors present the initial soil test of the location used. Authors can provide them if available Agreed and complied;Initial soil data was provided in the materials and methods section

In your statistical analysis, please confirm that the data followed normal distribution and equal variances, which justifies the use of ANOVA. 

The data is having normal distribution and equal variances, same is mentioned in the text also

In your results and tables, I will prefer you include the exact P values for the main effects and the interactions. This will make comparison easier. 

Agreed and complied; CD was already provided in the tables at 5% level of significance, now letters were also added to show the significant differences between treatment means

When reporting the specifics in the first few tables, you did not mention which planting year you are referring to. One can only generally assume that both years followed the same trend, but this was also not mentioned by the authors. 

Table 1, 6 and 7 data of both the years presented and in table 2 & 4, data of 2020; in table 3 & 5, data of 2021 was provided and same was mentioned in table title/heading.

---

## [Decision Letter · Decision Letter 1]

19 Jun 2023

Drip irrigation and sulphur fertilization influenced fodder yield, quality and water use efficiency of groundnut in arid region

PONE-D-23-01558R1

Dear Dr. Gautam

We’re pleased to inform you that your manuscript has been judged scientifically suitable for publication and will be formally accepted for publication once it meets all outstanding technical requirements.

Kind regards,

Arun Kumar Shanker

Academic Editor

PLOS ONE

Additional Editor Comments (optional):

Based on my evaluation of the revised manuscript and the author's response to the reviewers' suggestions, the authors have significantly improved the quality and presentation of their work.

Reviewers' comments:

Reviewer's Responses to Questions

**Comments to the Author**

1. If the authors have adequately addressed your comments raised in a previous round of review and you feel that this manuscript is now acceptable for publication, you may indicate that here to bypass the “Comments to the Author” section, enter your conflict of interest statement in the “Confidential to Editor” section, and submit your "Accept" recommendation.

Reviewer #1: All comments have been addressed

2. Is the manuscript technically sound, and do the data support the conclusions?

Reviewer #1: Yes

3. Has the statistical analysis been performed appropriately and rigorously? 

Reviewer #1: Yes

4. Have the authors made all data underlying the findings in their manuscript fully available?

Reviewer #1: Yes

5. Is the manuscript presented in an intelligible fashion and written in standard English?

Reviewer #1: Yes

6. Review Comments to the Author

Reviewer #1: All comments given to authors for improvements of the paper are addressed and it is acceptable now.

7. PLOS authors have the option to publish the peer review history of their article (what does this mean?). If published, this will include your full peer review and any attached files.

Reviewer #1: No

---

## [Editor Report · Acceptance letter]

26 Jul 2023

PONE-D-23-01558R1 

Drip irrigation and sulphur fertilization influenced fodder yield, quality and water use efficiency of groundnut in arid region 

Dear Dr. Gautam:

I'm pleased to inform you that your manuscript has been deemed suitable for publication in PLOS ONE. Congratulations! Your manuscript is now with our production department. 

Kind regards, 

on behalf of

Dr. Arun Kumar Shanker 

Academic Editor

PLOS ONE